

# Intra-rater and inter-rater reliability of the fixed plumb line for postural and scoliosis assessment in the sagittal plane: a pilot study

Federico Roggio[1,*], Bruno Trovato[1,*], Martina Sortino[1], Marta Zanghì[1], Claudio Di Brigida[2], Claudia Guglielmino[1], Claudia Lombardo[1], Carla Loreto[1], Piero Pavone[3] and Giuseppe Musumeci[1,4,5]

[1] Department of Biomedical and Biotechnological Sciences, Section of Anatomy, Histology and Movement Science, School of Medicine, University of Catania, Catania, Italy

[2] Catholic University of Murcia (UCAM), PhD program in Sports Science, Universidad de Murcia, Murcia, Spain

[3] Pediatric Clinic, Department of Clinical and Experimental Medicine, University Hospital A.U.O. ''Policlinico-Vittorio Emanuele'', University of Catania, Catania, Italy

[4] Research Center on Motor Activities (CRAM), University of Catania, Catania, Italy

[5] Department of Biology, Sbarro Institute for Cancer Research and Molecular Medicine, College of Science and Technology, Temple University, Philadelphia, United States of America

[*] These authors contributed equally to this work.

Corresponding author
Giuseppe Musumeci,
g.musumeci@unict.it

## ABSTRACT

**Background**. The plumb line (PL) is a common tool for assessing the sagittal curvatures of the spine, but its accuracy depends on the ability of the physician to use it correctly. This study aimed to present a fixed plumb line (FPL) no longer held by a physician but fixed to a support, evaluating the reliability in posture assessment, comparing it with PL in both adolescent with and without scoliosis.

**Methods**. The study evaluated the sagittal distances of the spine using a PL and a FPL in 80 young adults aged between $28.7 \pm 7.2$ and 55 adolescents aged between $12.4 \pm 2.3$, with and without scoliosis. Two expert and two novice clinicians tested the patients to evaluate the intra-rater and inter-rater reliability of FPL. Each clinician assessed participants twice on the same day, with a predetermined time interval ($>1$ h) to reduce recall bias. Multi-factor multivariate analysis of variance and two-way analysis of variance assessed the statistical significance, while intraclass correlation coefficient (ICC), standard error of measurement (SEM) and minimum detectable change (MDC) validated FPL consistency.

**Results**. FPL provided an ICC coefficient $>0.90$ for all the measures, while PL an average of 0.70. On AIS patients, PL and FPL showed a significant difference for C7 $p < 0.001$ and T12 $p < 0.001$. The measured parameters were sensitive to gender and age for the FPL, furthermore, the C7 and L3 measurements were statistically different between PL and FPL ($p < 0.001$). Intra-rater reliability results for FPL ranged from 0.94 to 0.98 across various parameters, while the SEM and MDC values underscore the valuable precision of the FPL with changes exceeding 1 cm being meaningful. These findings suggest that FPL could be a reliable and accurate tool for measuring sagittal distances of the spine in both scoliotic and non-scoliotic patients.

## INTRODUCTION

Suboptimal posture assessment methods can lead to misdiagnosis and ineffective treatment of conditions like posture alterations and scoliosis, potentially delaying treatment and negatively impacting patient outcomes. While digital methods offer greater precision (*Bottino et al., 2023*), the plumb line (PL) remains widely used due to its simplicity especially in orthopaedics. Introduced in 1855, the PL is a quick tool to quantify back posture (*Wilhelm Braune, 1985*). The PL is used by suspending a weight from a string to create a vertical reference line. For sagittal plane assessment, the patient stands naturally, and the plumb line aligns with specific spinal landmarks to help clinicians assess anterior-posterior balance and identify deviations like forward head posture, kyphosis, or lordosis. It is commonly used to measure the distance between the line and the 7th cervical, most prominent thoracic, 12th thoracic, 3rd lumbar, and 2nd sacral vertebrae. In the coronal plane, the PL passes over the spinous processes of the vertebrae. This allows for the detection of lateral deviations, which are indicative of scoliosis or other asymmetries in the trunk. However, the reliance of the PL on subjective clinician judgment creates variability (*Cohen et al., 2017*), highlighting the need for objective and reliable posture analysis methods. Its sensitivity in detecting scoliosis-related postural deviations in adolescents has been recognized for many years (*Romano et al., 2018*). Today, when used alongside other tests, the PL demonstrates high validity in identifying potential cases of adolescent idiopathic scoliosis (AIS) (*Grunstein et al., 2013*).

Scoliosis is a complex condition affecting 0.47–5.2% of the population (*Konieczny, Senyurt & Krauspe, 2013*; *Cheng et al., 2015*). It is characterized by the development of a structural, lateral, and rotated curvature of the spine in healthy children without other medical conditions around puberty. Factors influencing curve progression include bone maturity, curve size, and curve apex position. The diagnosis is made by exclusion, confirmed only after excluding other causes of scoliosis, such as vertebral malformations, neuromuscular disorders, and syndromic conditions (*Weinstein et al., 2008*). While standing coronal plane radiographs are essential for a definitive diagnosis and calculating the Cobb angle (*Knott et al., 2014*), various non-invasive methods exist to support the processes (*Torell, Nordwall & Nachemson, 1981*; *Bunnell, 1993*). These include rasterstereography (*Tabard-Fougère et al., 2017*), infrared thermography (*Roggio et al., 2023*), 3D ultrasound imaging (*Lai et al., 2021*) and other digital approaches (*Trovato et al., 2022*) all of which aid in monitoring scoliosis progression and guiding treatment decisions. Although rasterstereography (RS) is not a radiographic imaging method, it offers a digital approach with valid reliability for measuring sagittal distances (*Molinaro et al., 2022*). This non-invasive technique projects a grid pattern onto the back, capturing the surface contours and creating a 3D map of the spine. It is particularly useful for assessing scoliosis, as it measures spinal curvature and asymmetries without exposing patients to radiation (*Bassani et al., 2019*).

Recent research has broadened the applications of the PL, demonstrating its usefulness in AIS evaluation for patients undergoing lumbar surgery (*Zhang et al., 2020*), analyzing post-operative shoulder imbalance (*Berlin et al., 2022*), and assessing post-degenerative coronal imbalance (*Walker et al., 2021*). The measurements of trunk sagittal impairments taken with the PL have demonstrated their validity in the past. The PL has been effectively utilized as a supportive tool in AIS evaluation in patients who underwent lumbar surgery (*Zhang et al., 2020*), for the post-operative shoulder imbalance analysis (*Berlin et al., 2022*), and post-degenerative coronal imbalance assessment (*Walker et al., 2021*; *Higuchi et al., 2023*). It is considered a reliable source of information for clinicians in clinical setting, when evaluating people with scoliosis (*Grunstein et al., 2013*), providing guidance in both early screening and future decisions in scoliosis management (*Scaturro et al., 2021*).

While the PL test has benefits, it is subjective and depends on the accurate collection of the physician, as shown in Fig. 1A. If the clinician is not precise, it could lead to errors in the measurements and potentially bias the results (*Cohen et al., 2017*). Clinicians would benefit from having both hands free to collect measurements; however, the PL forces them to conduct measurements with one hand. An alternative, introduced in 2000, is the use of a laser plumb line projected over the back of the patient (*Zaina et al., 2007*). This method allows clinicians to perform hands-free evaluations. However, to date, there is a lack of studies employing this new method in different samples.

The absence of standardization in posture analysis can lead to significant variability in the results obtained. Therefore, it is vital to utilize objective methodologies that deliver accurate and reliable results. In response to this need, we propose an innovative method: the fixed plumb line (FPL). The FPL is a plumb line that has been fixed to a wall or roof using a support. In this way, the line is perpendicular to the ground and after stabilizing from any oscillations, it will remain still, making measurements easier and more precise on the sagittal plane, Fig. 1B. Therefore, the main difference between the PL and FPL is that the latter is held by a support, allowing the clinician to use both hands and thus achieve greater precision in measurements.

We hypothesize that the FPL might represent a notable advancement in this field of study, potentially reducing the subjectivity associated with traditional methods. Its secure attachment to a holder could eliminate the need for manual support or constant adjustments. Furthermore, we posit that the FPL will provide more consistent measurements over repeated trials compared to the traditional PL. However, it is important to consider the reliability of the FPL method. Therefore, this cross-sectional study has three primary aims: (1) to evaluate the inter-rater reliability of the FPL among young adults with four raters; (2) to evaluate the intra-rater in adolescents with and without scoliosis; (3) compare the FPL with the PL, and then compare both with RS.

## MATERIAL AND METHODS

This study involved two expert and two novice clinicians measuring sagittal spinal distances of 80 young adults using both PL and FPL. Additionally, 35 adolescents with AIS and 20 without, underwent the same measurements, complemented by RS analysis at the Research

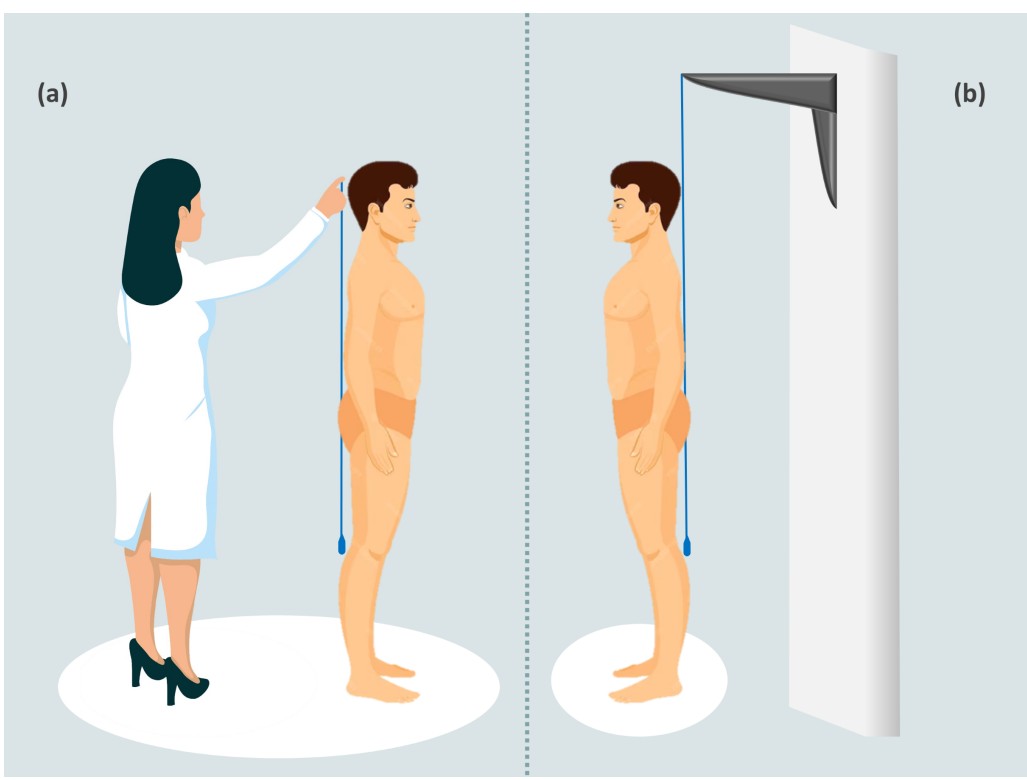

**Figure 1** **Classic plumb line *vs* fixed plumb line.** The figure shows two methods of measuring with the plumb line. In the first method, the classic execution of sagittal measurement with the plumb line is shown. This configuration can cause the plumb line to swing and move, resulting in inaccurate measurements (A). In the second method, the fixed plumb line is attached to the wall or roof with a support. This configuration ensures that the wire remains perpendicular to the ground and stable after the initial oscillations have stopped, making measurements easier and more accurate (B).

Center on Motor Activities (CRAM). Young adults were recruited at the Research Center on Motor Activities among sport science students. Adolescents with and without scoliosis were recruited from the Vittorio Emanuele General Hospital in Catania, recommended by the pediatrician during their first visit. All subjects were recruited from February 2022 to October 2022. Inclusion criteria included: patients recruited at their first visit, age range "adolescents" according to SOSORT guidelines (10–17 years), *Negrini et al. (2018)*, both genders. Exclusion criteria included: back pain, diagnosed neuromuscular disease, Scheuermann's disease or thoracic kyphosis exceeding 60 Cobb degrees, spondylolisthesis or other spinal deformities, prior treatment or use of brace. The minimum sample size of 50 was determined a priori using GPower 3.1 software with an effect size of 0.5 and power of 0.95. The study was approved by the Research Center on Motor Activities (CRAM) Scientific Committee—University of Catania (Protocol n.: CRAM-019-2021, 20/12/2021) and adhered to the Declaration of Helsinki. Young adults and the parents of adolescent participants signed written informed consent forms to agree to participate in the study.

### Inter-rater data collection

To minimize order effects, we randomized posture assessment tools (PL and FPL) using an ABBA counterbalancing method. Four clinicians performed the measurements, each assessing participants twice within the same day with a predetermined time interval of over one hour to reduce recall bias. Each rater was unaware of the measurements performed by the others. Measurements were taken with the participant in the anatomical zero position described by *Kendall et al. (1993)*. For the PL measurements, the physician held the PL at the level of the occipital prominence and palpated to locate the C7 spinous process (*Póvoa et al., 2018*). Then, the physician individuated the 8th thoracic vertebra and palpated to determine the T12 position and the kyphosis apex. The L3 process was identified as the midpoint between the posterior iliac crests; additionally, the physician also identified the S2 reference point. The sagittal imbalance was calculated by subtracting the measurement of S2 from the measurement of C7. All measurements were taken with a rigid millimeter ruler and rounded to the nearest 0 or 5 units. For the FPL measurements, the plumb line was attached to a roof support and left free to swing until it stopped. Then, we collected the same parameters of the PL. In order to compare the PL and FPL results, the distance of the most prominent thoracic point was subtracted from all measurements to obtain the relative sagittal distances.

### Intra-rater scoliosis data collection

Adolescents with and without scoliosis underwent the procedure described before. Measurements were taken randomly from 3 PM to 6 PM. Participants returned after 1 week for intra-rater reliability assessment with FPL. Participants were grouped based on Cobb angle: no scoliosis (NS), mild scoliosis (MS) < 20 degrees, and moderate scoliosis (MoS) 21–35 degrees. After collecting sagittal distances with both the PL and FPL, participants underwent RS analysis using the Spine 3D system (Sensormedica, Rome, Italy). This method reconstructs the back digitally and measures various parameters, but we considered only the distance between a digital line and the 7th cervical (C7) and 3rd lumbar (L3) vertebrae. The participants were asked to stand still on a designated area and to face the camera of the Spine 3D with their back naked. This system is marker-less and does not require any intervention from the clinician, allowing an objective evaluation free from human error.

### Statistical analysis

Data analysis was conducted with R Project software (Vienna, Austria). We tested data normality with Shapiro–Wilk and homogeneity of variances with Bartlett's test. For the first part of the study, we used the intraclass correlation coefficient (ICC2k), along with Fisher's Z-transformation to test for a significant difference between the ICCs of the two methods. Next, a multi-factor MANOVA assessed PL and FPL measurements considering scoliosis severity, curve, gender, and age. A two-way ANOVA compared PL and FPL by scoliosis severity, and a second two-way ANOVA compared C7 and L3 across methods (PL, FPL, RS). Significant differences ($p < 0.05$) were identified using Tukey's HSD. Finally, the intra-rater reliability of the FPL in AIS patients was further assessed with ICC, minimum detectable change (MDC), and standard error of measurement (SEM).
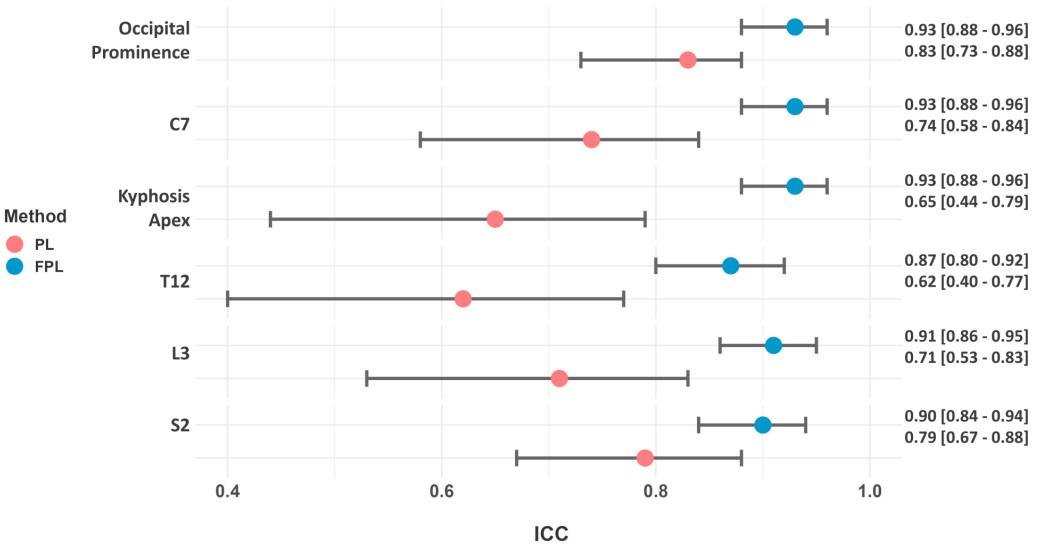

**Figure 2** Forrest plot of the ICCs values of both PL and FPL when measured by four different raters.

**Table 1** Characteristics of the participants.

| Participants | Age (years) | | Weight (kg) | | Height (cm) | |
|---|---|---|---|---|---|---|
| | Mean | SD | Mean | SD | Mean | SD |
| NS | 13.4 | 2.0 | 50.2 | 12.7 | 155.5 | 12.2 |
| MS | 13.2 | 2.5 | 49.0 | 13.9 | 156.1 | 13.1 |
| MoS | 12.5 | 2.4 | 47.3 | 13.9 | 152.7 | 15.1 |

**Notes.**
NS, no scoliosis; MS, mild scoliosis; MoS, moderate scoliosis.

## RESULTS

The characteristics of the participants of the first part of the study are: mean age of $28.7 \pm 7.2$ years, mean weight of $62.8 \pm 7.8$ kg, mean height of $169.9 \pm 7.2$ cm. The ICCs of the FPL were constantly higher than the PL, as illustrated in Fig. 2. Furthermore, the Fisher's Z-test confirmed a statistically significant difference in precision between the two methods (occipital prominence $p = 0.005$, C7 $p < 0.001$, kyphosis apex $p < 0.001$, T12 $p = 0.0003$, L3 $p = 0.0001$, S2 $= 0.01$).

After we ensured the reliability of the FPL across raters, we conducted the second part of the study on adolescent with and without scoliosis, whose characteristics are reported in Table 1. The Risser stage median value was MS $= 2$ and MoS $= 1$, the mean Cobb angle was MS $= 14.49° \pm 2.90°$ and MoS $= 27.23° \pm 3.92°$ and the type of curve was MS $= 9$ C *vs* 10 S shape, MoS $= 11$ C *vs* 4 S shape. Table 2 shows the mean and standard deviation of PL and FPL.

The first step was to perform MANOVA analysis for the PL and FPL measurements separately and then compare the results of the two methods. As shown in Table 3, the PL method only showed weak differences for C7, while no valuable differences were found for

**Table 2  Mean values in millimetres of the landmarks of the spine.**

|  | NS | | MS | | MoS | |
|---|---|---|---|---|---|---|
|  | **Mean** | **SD** | **Mean** | **SD** | **Mean** | **SD** |
| PL |  |  |  |  |  |  |
| C7 | 16.5 | 11.1 | 6.5 | 6.7 | 6.6 | 12.1 |
| T12 | 26.7 | 16.0 | 12.7 | 16.3 | 10.6 | 26.4 |
| L3 | 39.6 | 12.3 | 29.5 | 8.0 | 32.3 | 20.8 |
| S2 | 16.4 | 9.5 | 16.0 | 12.0 | 17.0 | 13.3 |
| FPL |  |  |  |  |  |  |
| Occipital prominence | 21.2 | 27.1 | 6.9 | 25.6 | 13.1 | 27.4 |
| C7 | 42.1 | 19.0 | 43.4 | 23.8 | 32.7 | 20.9 |
| T12 | 38.8 | 14.9 | 38.8 | 16.2 | 36.9 | 21.5 |
| L3 | 39.7 | 16.7 | 40.6 | 22.2 | 37.7 | 20.3 |
| S2 | 27.6 | 23.7 | 21.5 | 19.6 | 18.1 | 19.8 |

Notes.
NS, no scoliosis; MS, mild scoliosis; MoS, moderate scoliosis; SD, standard deviation.

**Table 3  Multi-factor MANOVA p-values of plumb line and fixed plumb line separately.**

| Dependent variable | Gender | Age | Scoliosis severity | Scoliosis curve | Gender × age |
|---|---|---|---|---|---|
| PL |  |  |  |  |  |
| C7 | 0.677 | 0.571 | 0.067 | 0.865 | 0.087 |
| Kyphosis apex | 0.868 | 0.328 | 0.309 | 0.486 | 0.298 |
| T12 | 0.366 | 0.643 | 0.357 | 0.251 | 0.856 |
| L3 | 0.918 | 0.449 | 0.702 | 0.230 | 0.151 |
| S2 | 0.825 | 0.810 | 0.799 | 0.922 | 0.232 |
| FPL |  |  |  |  |  |
| Occipital prominence | 0.744 | 0.071 | 0.328 | 0.781 | 0.977 |
| C7 | 0.08 | 0.016[*] | 0.387 | 0.659 | 0.208 |
| Kyphosis apex | 0.017[*] | 0.41 | 0.487 | 0.064 | 0.012[*] |
| T12 | 0.049[*] | 0.041[*] | 0.781 | 0.937 | 0.006[*] |
| L3 | 0.205 | 0.640 | 0.818 | 0.226 | 0.274 |
| S2 | 0.034[*] | 0.241 | 0.423 | 0.571 | 0.554 |

Notes.
PL, plumb line; FPL, fixed plumb line.
[*]$p < 0.05$.

the other parameters. In the FPL method, significant differences were found for Occipital prominence, C7, Kyphosis apex, T12, L3 and S2, Table 3. The interaction effect of scoliosis severity and curve were not statistically significant. We then compared the PL and FPL measurements finding significant differences between the two methods for C7 ($F = 44.891$, $p < 0.001$), T12 ($F = 25.161$, $p < 0.001$). There was a weak difference for L3 ($F = 3.801$, $p = 0.098$) and no differences for Kyphosis apex and S2. The analysis for the scoliosis severity or PL/FPL × scoliosis severity did not provide any significant difference (Table 4).

**Table 4 Two-way ANOVA results between the landmarks of the spine.**

|  | C7 | Kyphosis apex | T12 | L3 | S2 |
|---|---|---|---|---|---|
| PL/FPL | <0.001[**] | 0.197 | <0.001[**] | 0.098 | 0.150 |
| Scoliosis severity | 0.144 | 0.449 | 0.389 | 0.824 | 0.614 |
| PL/FPL × scoliosis severity | 0.401 | 0.790 | 0.281 | 0.852 | 0.542 |

Notes.
 PL, plumb line; FPL, fixed plumb line.
[**] $p < 0.001$.

**Table 5 Results of the intra-rater reliability of the FPL.**

|  | ICC | CI | SEM | SEM (mm) | MDC | MDC (mm) |
|---|---|---|---|---|---|---|
| Occipital prominence | 0.98 | 0.96–0.99 | 0.72 | 4 | 2.00 | 11 |
| C7 | 0.97 | 0.94–0.98 | 0.92 | 3 | 2.56 | 7 |
| Kyphosis apex | 0.97 | 0.95–0.97 | 1.01 | 4 | 2.81 | 10 |
| T12 | 0.95 | 0.92–0.97 | 1.31 | 4 | 3.65 | 11 |
| L3 | 0.95 | 0.91–0.97 | 1.30 | 4 | 3.61 | 10 |
| S2 | 0.94 | 0.90–0.97 | 1.29 | 4 | 3.58 | 10 |

Notes.
 ICC, intra-class correlation 2,k; CI, confidence interval; SEM, standard error of measurement; MDC, minimum detectable change.

Finally, we compared the C7 and L3 measurements between PL, FPL and RS. Two-way ANOVA analysis showed a significant difference in C7 with $p < 0.001$ and L3 with $p = 0.048$ when controlled between the different methods, and the Tukey HSD test revealed significant differences between PL and FPL ($p < 0.001$) and PL and RS ($p < 0.001$), but no differences between FPL and RS ($p = 0.631$).

### Reliability of the fixed plumb line

The ICC (2,k) results reported in Table 5, revealed excellent reliability across all measurements, as indicated by the high values ranging from 0.94 to 0.98 (Fig. 3). Furthermore, based on these findings, we calculated the actual SEM and MDC values, rounded to the nearest integer, as shown in Table 5 as SEM (mm) and MDC (mm). These findings demonstrate the high consistency and reproducibility of the FPL measurements over time.

## DISCUSSION

Evidence supporting routine scoliosis screening varies globally, being mandatory in Japan but not legislated in the United States or Europe (*Sabirin et al., 2010*; *Altaf et al., 2017*). Research since the 1960s has demonstrated that early detection of scoliosis is effective in preventing its progression (*Grivas et al., 2007*). Yet, without screening, scoliosis may remain unnoticed, as seen in 36% of Italian adolescents with clinical signs (*Trovato et al., 2016*).

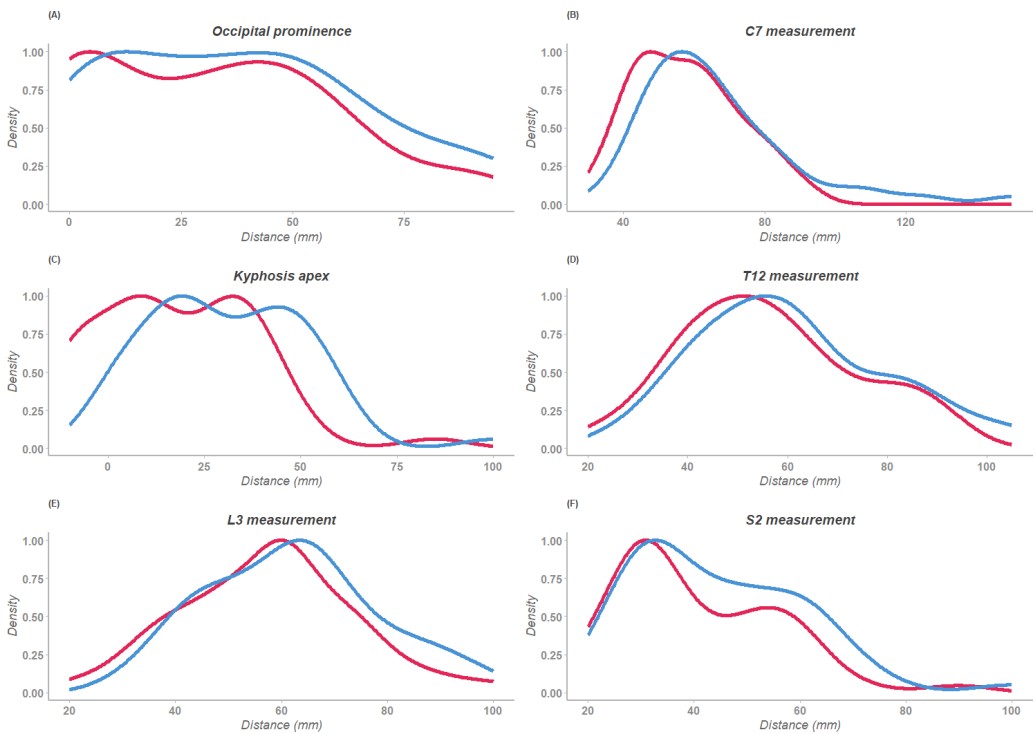

**Figure 3** **Density plots of the sagittal measurements collected twice.** The red line represents the first collection; the blue line represents the second collection after one week. (A) Occipital prominence, (B) C7, (C) Kyphosis apex, (D) T12, (E) L3, (F) S2.

Specifically, early detection and accurate assessment are crucial for effective management and treatment of this spinal deformity, which can prevent the progression of this condition and reduce the need for more invasive treatments later (*Richards & Vitale, 2008*). The initial clinical assessment plays a vital role in identifying potential cases of scoliosis, determining the severity of the condition, and guiding further diagnostic procedures. Among the various tools available for clinical assessment, the PL stands out for its simplicity, portability, and effectiveness. Non-invasive screening methods, such as the use of the PL, RS, or infrared thermography, pose no risk to the patient, making them ideal for initial screening (*Horne, Flannery & Usman, 2014*) or school scoliosis screening (*Kwok et al., 2017*).

*Zaina et al. (2012)* conducted a study focusing on the importance of sagittal plane measures in managing AIS and hyperkyphosis. By recruiting 180 adolescents, they reported normative data and measurement errors of the PL. However, they only reported the C7 and L3 parameters. Similarly, the study by *Negrini et al. (2019)* did so for 584 healthy individuals, reporting the following sagittal results: C7: 39.9 ± 16.7 mm, T12: 21.4 ± 15.3 mm, L3: 39.9 ± 15 mm, and S2: 20.6 ± 17.0 mm. Some similarities were present with our results only for T12: 26.7 ± 16.0 mm, and L3: 39.6 ± 12.3 mm. Despite having valid samples, their results and ours differ, likely due to the technique of the physician influencing accuracy, as *Grosso et al. (2002)* found when noting lower reliability in certain vertebral measurements between observers. Therefore, the great variability in the use of

the PL can be attributed to several factors that complicate its precise application. One significant source of variability is the difficulty clinicians face in keeping the PL still at a precise point. The inherent movement and oscillation of the PL, or of the patient, make it challenging to maintain a consistent vertical reference. Additionally, clinicians must hold the PL steady with one hand while simultaneously measuring distances and recording the results with the other hand. This dual-task requirement increases the likelihood of measurement errors and inconsistencies, as even slight movements or misalignments can significantly impact the accuracy of the recorded distances. Consequently, these practical challenges highlight the necessity for more reliable and user-friendly methods, such as the FPL, which can reduce variability and enhance measurement precision.

For the inter-rater reliability, the FPL method demonstrated an excellent inter-rater reliability with an ICC > 0.90 for almost all parameters on the sagittal plane except for the T12 parameter. In contrast, the PL method had a mean ICC of 0.70, indicating moderate reliability. These results suggest that the FPL method is a more precise measurement technique than the PL method. The high ICC of the FPL method across four raters with varying experience indicates that the FPL is a technique that can be easily disseminated among different practitioners and clinicians. This could facilitate standardized comparisons of postural measures between different locations, addressing a limitation of the PL method due to its reduced reproducibility. Reproducibility is a critical factor in clinical measurements as it ensures that different practitioners can achieve similar results under consistent conditions (*Weir, 2005*). One of the main causes of this difference is that the PL method is held with the hand of the operator, while the FPL is fixed to the wall with a support, thus reducing operator-related variability. By eliminating the need for the clinician to hold the plumb line, the FPL method minimizes these sources of error, resulting in more consistent and reliable measurements.

For the intra-rater reliability, we evaluated the consistency of the FPL method, focusing on ICC, SEM, and MDC. Our results showed excellent reliability for all measurements, with ICC values ranging from 0.94 to 0.98, indicating a high level of agreement between repeated measurements and affirming the robustness of the FPL method in clinical assessments. The SEM and MDC provided significant insights into measurement precision and variability. All calculated SEM values were approximately 4 mm, suggesting that any single measurement taken with the FPL method will likely deviate by only a small margin from the true value, which is crucial for maintaining consistency in clinical practice. The MDC values, which ranged from 7 mm to 11 mm for all parameters, underscore that changes in measurements exceeding about 1 cm are meaningful and not merely a result of measurement variability. This threshold is vital for clinicians to distinguish between actual changes in sagittal alignment and those within the expected measurement error. When the sagittal measures of a patient change by more than the MDC value, clinicians can be confident that this change is significant and not due to chance. The high ICC values, combined with the low SEM and practical MDC thresholds, support the FPL method as a precise and reliable instrument for monitoring sagittal alignment. This reliability is particularly important in clinical settings where consistent and accurate measurements are critical for tracking the progression of spinal deformities and adjusting treatment plans

accordingly. The ability of the FPL to provide consistent measurements across different raters and repeated assessments on the same day reinforces its utility in various clinical environments.

Building on these findings, we finally analyzed the statistical differences between PL and FPL methods when used with AIS patients. PL measurements showed a minor influence from scoliosis curve and gender at C7. In contrast, FPL measurements were significantly affected by gender and age, especially at T12 and the Kyphosis apex, while scoliosis severity and curve seem to have no substantial effect on either method. The FPL likely provided more accurate results due to better methodology, demonstrating strong correlations with gender and age at T12. Comparisons indicated significant differences in C7 and T12 measurements between PL and FPL, but none between FPL and RS, highlighting the methodological advantages of FPL. Thus, the lack of statistical differences between RS and FPL can be attributed to the good reproducibility and reliability of the method, considering that RS is already recognized as a reliable non-invasive technique that can provide a radiation-free alternative (*Mehta et al., 2023*). Although RS was not the main objective of this study, we included it as other studies have assessed its validity as a non-invasive method in scoliosis screening. The primary advantage of RS is its ability to provide quick, accurate, and detailed information about spinal alignment without exposing the patient to ionizing radiation, a significant concern with traditional radiographic methods. This makes RS particularly suitable for repeated assessments, which are often necessary in the monitoring and management of scoliosis, especially in children and adolescents (*Mohokum, Schülein & Skwara, 2015*). Clinical studies have demonstrated that RS cannot be considered a valid alternative to radiographic evaluation (*Bassani et al., 2019*). However, it is valid in measuring parameters such as the curvature of the spine in the coronal and sagittal planes, providing valuable data that can aid in the diagnosis and treatment planning for scoliosis (*Bassani et al., 2019*). Due to this consideration, we did not consider RS as the gold standard for the study. Instead, we included its C7 and L3 parameters to strengthen the consistency of the FPL measurements, as RS still demonstrates good validity for sagittal parameters (*Degenhardt et al., 2017*; *Degenhardt, Starks & Bhatia, 2020*).

Lateral projection X-rays represent the gold standard for assessing conditions such as scoliosis and hyperkyphosis (*Negrini et al., 2019*). Despite their widespread use, the radiation exposure is a significant limitation. While a single exposure to ionizing radiation may not be harmful, adolescents with AIS are exposed to repetitive radiations due to follow-up X-rays, which can be harmful during growth (*Himmetoglu et al., 2015*). Low-dose X-ray systems, such as the EOS imaging system, have been developed to reduce radiation exposure while maintaining diagnostic accuracy (*Hui et al., 2016*). Studies have shown that low-dose X-rays can achieve significant radiation dose reductions compared to conventional X-rays (*Deschênes et al., 2010*). This reduction is particularly beneficial for children and adolescents, who are more susceptible to the harmful effects of ionizing radiation (*Cheng et al., 2015*). However, due to the limited availability of the EOS system, as not all hospitals may have it, and since X-rays are typically prescribed only when scoliosis is suspected, clinicians need valid and comparable tools like the PL for initial assessments. The PL is not the only valid tool, but its portability, user-friendliness, and

speed make it an excellent resource for clinicians working in postural assessment and scoliosis screening. This study presented the FPL, which shows good reproducibility in measuring sagittal distances of the spine in both normal subjects and those with scoliosis. However, we cannot ensure its validity because, to truly determine if the sagittal distances are representative of spinal misalignments, these measurements should be compared with sagittal X-rays in a comparative study. To date, the only study that attempted this is by *Negrini et al. (2019)*, where the authors compared only the Sagittal Index, *i.e.,* the sum of C7 and L3 measures, between the PL and X-rays. A comparative study could provide a standard error to consider when measuring these parameters with a PL, thereby offering a more accurate representation of the sagittal distances of the considered anatomical landmarks.

While statistical evidence does not always guarantee clinical relevance, we believe the FPL will improve clinical practice by offering a reliable method that reduces operator error. There are some limitations to this study. First, a selection bias exists due to the pre-screening of participants for scoliosis; second, we could not compare the FPL measures with sagittal X-ray measures as we had only the coronal image. Comparing our results with the radiographs could have allowed the evaluation of the error between FPL and X-rays, giving more practical information to clinicians, and allow a calculation of the accuracy of the FPL method. Therefore, further studies are needed to validate the FPL through comparison with sagittal X-rays, expand its use across a wider age range, and employ a larger sample size in the general population.

## CONCLUSIONS

The findings of this study demonstrated that the FPL method yields reliable and precise measurements for spinal alignment in individuals with and without scoliosis and is sensitive to variations in gender and age at key spinal points of scoliosis. (1) Regarding the inter-rater reliability of the FPL, the results showed excellent ICC values (>0.90), suggesting that this could be a valuable tool for clinicians assessing sagittal spinal alignment among healthy participants. (2) For the intra-rater reliability of the FPL, the combination of high ICC values (>0.94), low SEM, and reasonable MDC values demonstrates that the FPL method is a highly reliable and precise tool for assessing spinal alignments in the sagittal plane. (3) Finally, by comparing the results of the PL, FPL, and RS, the study highlighted that the measurements of the FPL were more consistent than those of the PL when compared to RS, which is an automatic and clinician-bias-free method.

Therefore, this study supports the FPL as a reliable tool for measuring sagittal distances of the spine in individuals with and without scoliosis.

### Funding

This work was supported by the University Research Project Grant (PIACERI Found—PreRoF4 OA—2024–2026), Department of Biomedical and Biotechnological Sciences

(BIOMETEC), University of Catania, Italy. The funders had no role in study design, data collection and analysis, decision to publish, or preparation of the manuscript.

### Grant Disclosures
The following grant information was disclosed by the authors:
University Research Project: (PIACERI Found—PreRoF4 OA—2024–2026).
Department of Biomedical and Biotechnological Sciences (BIOMETEC), University of Catania, Italy.

### Competing Interests
The authors declare there are no competing interests.

### Author Contributions
- Federico Roggio conceived and designed the experiments, performed the experiments, analyzed the data, prepared figures and/or tables, and approved the final draft.
- Bruno Trovato conceived and designed the experiments, performed the experiments, prepared figures and/or tables, and approved the final draft.
- Martina Sortino conceived and designed the experiments, prepared figures and/or tables, and approved the final draft.
- Marta Zanghì conceived and designed the experiments, prepared figures and/or tables, and approved the final draft.
- Claudio Di Brigida conceived and designed the experiments, authored or reviewed drafts of the article, and approved the final draft.
- Claudia Guglielmino performed the experiments, analyzed the data, authored or reviewed drafts of the article, recruiting participants, and approved the final draft.
- Claudia Lombardo performed the experiments, authored or reviewed drafts of the article, and approved the final draft.
- Carla Loreto conceived and designed the experiments, authored or reviewed drafts of the article, and approved the final draft.
- Piero Pavone performed the experiments, authored or reviewed drafts of the article, experiment supervision, and approved the final draft.
- Giuseppe Musumeci conceived and designed the experiments, authored or reviewed drafts of the article, funding acquisition, and approved the final draft.

### Human Ethics
The following information was supplied relating to ethical approvals (*i.e.*, approving body and any reference numbers):

Research Center on Motor Activities Scientific Committee.

### Data Availability
The raw measurements are available in the Supplementary File.

## Supplemental Information

Supplemental information for this article can be found online at http://dx.doi.org/10.7717/peerj.18121#supplemental-information.

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
