# Peer review of "Intra-rater and inter-rater reliability of the fixed plumb line for postural and scoliosis assessment in the sagittal plane: a pilot study"

_PeerJ, doi:10.7717/peerj.18121_

## Round 0.1 · original submission · Major Revisions

I agree with the comments of the reviewers in that there should be some more details added to the methods with respect to the two conditions and specifically the calculation of the SEM and MDC. The discussion is a little short and underwhelming - the reviewers both suggest ways in which this could be improved. Please also comment on where you think the difference between the methods comes from - is this the error of the physician keeping the PL in the correct place? Also refer back to your original aims to show that these were satisfied.

Please specifically describe what the values in table 3 are in the caption (p-values?).

Reviewer 1 ·

Basic reporting

Authors compared PL and FPL in 80 young adult and 55 adolescents with and without scoliosis, and reported that FPL is reliable and accurate.

Experimental design

fine

Validity of the findings

fine

Additional comments

As PL can be either sagittal and coronal, please indicate that this article is regarding sagittal PL including the title. Please add the number of observers in the method section of the abstract. Much more detail is needed to explain the method of PL and FPL in addition to figure 1. As authors mentioned in the limitation section, accuracy cannot be reported based on this paper as it was not compared with sagittal x-rays including EOS. Please discuss more extensively regarding that issue.

·

Basic reporting

Abstract:
Considering that the abstract could contain up to 500 words, according to the journal's guidelines, I suggest including the established age for adolescents and young adults. Additionally, include how many people conducted the assessments and the time interval between the assessments.
Objective: This study aimed to present a fixed plumb line (FPL), no longer held by a physician but fixed to a support.
- I suggest rewriting the objective to include "test the intra- and inter-rater reproducibility of the plumb line distances in the sagittal plane."

Introduction:
The authors argue that the use of the plumb line requires training and expertise and that its handling could affect the results. In this context, the authors suggest the use of a fixed plumb line, reducing bias during collection. My question is whether the authors could also discuss the use of a laser plumb line, which would also be a way to keep the plumb line fixed, allowing the evaluator to remain hands-free.
Moreover, it is necessary to better explore in the introduction the measures obtained with the plumb line and their importance in the assessment, monitoring, and decision-making regarding patients with scoliosis.
At the end of the introduction, three different objectives are presented: 1) inter-rater reliability of the FPL comparing an expert with a novice in young adults, 2) FPL vs. PL and this vs. topography??? 3) intra-rater reliability of the FPL in adolescents with and without scoliosis.
The objective is not consistent with the introduction and does not clearly outline the rationale of the study.

Experimental design

Materials and Methods:
Could the authors provide more details on how and where the patients were recruited?
Is it possible to describe in more detail whether the evaluation was the first one the patients underwent, or if assessments during the follow-up of the patients were also included?
Were the patients receiving any type of treatment? Did they use a brace? Was the collection done on the same day as the physiotherapy appointment, for example? For those who used a brace, how long did they remain without it before the evaluation? How long had these patients been treated?
Could the authors clarify the eligibility criteria for the age and sex of the subjects included?
What was the minimum sample size obtained through the sample size calculation performed in G*Power? I suggest including this information in the text.
Were the distances collected with a measuring tape and plumb line all in the sagittal plane? Measuring the distances at C7, T8, T12, L3, and S2, correct? From these five distances, sagittal balance was extracted; I ask: is there a reference value for each of the collected distances and the calculated parameter?
The authors need to describe in detail how the surface topography was collected and which parameters were extracted. Were the surface topography collection and analysis performed by the same evaluators who used the plumb line?
Could the authors describe whether there was any blinding of the results between evaluators and between instruments?
In the "Scoliosis data collection" section, what do the authors mean by "C7 and L3 measurements were collected"? What parameters are extracted from these measurements? And how is this related to scoliosis?

Validity of the findings

Results:
The presentation of ICCs with a forest plot is interesting.
Why does Table 2 not report data on the curve's apex?
Do the authors believe that the differences found are due to greater data variability when using the PL and lesser variability when using the FPL, allowing differentiation of situations beyond error?

Discussion:
There are missing elements about surface topography and its comparison with clinical assessment.
Is an MDC of 1 cm considered good for this measure?
At the end of the discussion, the authors state that one of the study's limitations is the lack of comparison with X-rays in the sagittal plane and that new studies should explore this relationship. Could the authors provide examples of how these comparisons would be made and their clinical significance?

Conclusion:
I suggest adding in the conclusion and throughout the text that the plumb line measurements are exclusively in the sagittal plane. I also suggest organizing the conclusion by separately addressing each of the outlined objectives: 1) inter-rater reliability (expert vs. novice); 2) FPL vs. PL and this vs. topography???; and 3) intra-rater reliability of the FPL in adolescents with and without scoliosis.

Additional comments

I appreciate the opportunity to review this manuscript. The study is interesting and provides a practical perspective on the use of a simple, low-cost, and easy-to-handle assessment tool. However, the introduction needs more in-depth points regarding the studied population, the assessment tools used, the parameters to be extracted, and their clinical applicability. The objective in the abstract and at the end of the introduction are different; I suggest unifying them more clearly and precisely. I suggest major revisions.

---

## Round 0.2 · accepted · Accept

Congratulations on your acceptance. Please double-check the title and amend accordingly during production. Intra-rater is referred to twice (one instance should be 'inter').

Reviewer 1 ·

Basic reporting

The authors successfully addressed all my comments. Now the quality of the paper improved a lot. I appreciate it.

Experimental design

s/a

Validity of the findings

s/a

Additional comments

s/a

·

Basic reporting

No comment.

Experimental design

No comment.

Validity of the findings

No comment.

Additional comments

The authors did a good job revising the manuscript according to the suggestions provided. The manuscript is now clearer and includes important methodological points. I only suggest a minor revision in the title. Please review the title; there was probably a typo: Inter- and intra-rater reliability of the fixed plumb line for postural and scoliosis assessment in the sagittal plane: a pilot study.